# Endodontic Management of Endo-Perio Lesions

Bogdan R Shumilovich [1], Vladimir V Rostovtsev [1], Vadim A Kunin [1], Irina S Bishtova [1], Russell A. Paul [2], Igor Tsesis [2] and Dan Littner [2],*

[1] Department of Postgraduate Dentistry, Voronezh N.N. Burdenko State Medical University, 394036 Voronezh, Russia; bogdanshum@gmail.com (B.R.S.); rosvrn@gmail.com (V.V.R.); kunin.911@rambler.ru (V.A.K.); bishtova.ira@mail.ru (I.S.B.)

[2] Department of Endodontology, Maurice and Gabriela Goldschleger, School of Dental Medicine, Tel Aviv University, P.O. Box 39040, Tel Aviv 6997801, Israel; russpaul07@gmail.com (R.A.P.); dr.tsesis@gmail.com (I.T.)

* Correspondence: littnerdan@gmail.com; Tel.: +972-3-642-1397

**Abstract:** An endo-perio lesion is one of the more common tooth-related problems. An association between the presence of apical and marginal periodontitis is known in the literature and has been observed in 5.7% of individuals aged 40–45 years old. The purpose of the present article is to present three case reports describing the successful retreatment of endo-perio lesions. In each of these cases, we used a biologically active bioceramic root canal sealer, GuttaFlow Bioseal, which is a bioactive root canal filling material composed of gutta percha, polydimethylsiloxane, platinum catalyzer, zirconium dioxide, and bioglass. All cases were followed up clinically and radiographically for a period of at least 11 months.

**Keywords:** endodontic-periodontal lesions; endodontic retreatment; bioceramic sealer





## 1. Introduction

Endodontic-periodontal lesions, which are primarily endodontic and secondarily periodontal in etiology, are a commonly encountered dental condition. An association between the presence of apical and marginal periodontitis is known in the literature and has been observed in 5.7% of individuals aged 40–45 years old [1,2]. They may be due to the close anatomical association between endodontic and periodontal tissues, which allows for infections to spread between the root canal and marginal periodontium [1,3]. A significant association between the presence of apical and marginal periodontitis has been reported [4].

Endodontic-periodontal lesions present challenges to clinicians with respect to diagnosis, treatment, and prognosis of the teeth involved [5,6].

Diagnosis is complicated by the fact that these diseases are too frequently viewed as independent entities. Misdiagnosis and subsequent incorrect treatment choices may then ultimately necessitate tooth extraction.

According to the traditional classifications of Simon et al. [7], primary endodontic lesions manifest themselves clinically with possible drainage from the gingival sulcus and swelling in the attached gingiva. A necrotic pulp may be associated with a sinus tract extending from the apex along the root surface and exiting at the cervical line. In multi-rooted teeth, the sinus tract may drain into the bifurcation area, with associated radiographic evidence of periodontal involvement (bone loss, appearing as a radiolucency along the root).

An accumulation of plaque at the gingival margin may eventually result in marginal periodontitis, and if the primary endodontic disease remains untreated, there may be secondary periodontal destruction. Simon termed this condition primary endodontic lesions with secondary periodontal involvement [7,8].

However, there remains significant controversy concerning the traditional classification schemes and as to how these conditions should be further subdivided into additional subgroups as the pathology progresses.

Tsesis et al. [5] suggested the use of a three-component categorization scheme of endodontic-periodontal lesions, based on the primary etiological factor of the pathology and clinical presentation:

1.  Purely endodontic lesion: when the pulp is necrotic and infected and there is a draining sinus tract coronally through the periodontal ligament into the gingival sulcus.
2.  Purely periodontal lesion: when a deep periodontal lesion involves most of the root surface and the dental pulp is vital.
3.  Endodontic-periodontal lesion: when the pulp is necrotic and infected and there is a deep periodontal pocket.

When an endo-perio lesion is diagnosed, both endodontic and periodontal therapy may be required. In this case, the tooth prognosis depends mainly on the success of the endodontic treatment, followed by appropriate supplemental periodontal therapy.

The aim of this study was to present clinical cases of endo-perio lesions that were successfully treated using a modern endodontic technique.

## 2. Materials and Methods

### 2.1. Clinical Case 1

A healthy 49-year-old Caucasian male presented at the clinic with a chief complaint of persistent pain and discomfort in his second left mandibular molar. The patient reported previous endodontic treatment 7 years ago.

**Intra-oral clinical examination**

The second left mandibular molar had been restored with composite resin restoration. The tooth exhibited tenderness on percussion, with a probing depth within normal limits (<3 mm), except for the distal surface of the tooth, where it was 7 mm.

**Radiographic examination**

Preoperative cone-beam computed tomography (CBCT) imaging revealed previous root canal obturation, with a large radiolucent area in the distal aspect of the tooth. Furthermore, there was considerable resorption of the buccal and lingual bone plate on the distal aspect of the tooth (Figure 1a,b). The CBCT axial view revealed a C-shape root canal configuration.

**Endodontic Diagnosis**

Pulpal diagnosis—previously treated

Periapical diagnosis—symptomatic apical periodontitis

Primary endodontic—secondary periodontal lesion

**Periodontal Diagnosis**

Localized periodontitis, Stage III (according to the classification which was presented at the 2017 World Workshop by the American Academy of Periodontology (AAP) and the European Federation of Periodontology (EFP) [9].

**Treatment**

After discussing treatment options with the patient, a decision was made in favor of endodontic intervention as the first stage of a comprehensive treatment plan, followed by periodontal treatment.

Treatment was performed under magnification with a dental operating microscope. After local anesthesia (2% lidocaine with 1:100,000 epinephrine) and rubber dam isolation, the old restoration was removed and the appropriate access cavity was refined, which revealed a C-shape root canal configuration: Category II (C2) according to Fan's anatomic classification (Figure 1c).

Cleaning and shaping of the root canal system was performed using HyFlex EDM (Coltene, Switzerland) with abundant 5% sodium hypochlorite and 17% EDTA irrigation (Figure 1d).

The working length was determined by an electronic apex locator (Propex II; Dentsply Maillefer, Ballaigues, Switzerland).

Root canal obturation was performed with the GuttaFlow Bioseal system (Coltene, Roeko, Germany) (Figure 1e). The endodontic access cavity was restored with the composite material, and a temporary filling with a monolithic zirconia crown was applied. Periodontal therapy followed the endodontic treatment as soon as possible.

Postoperative CBCT showed complete root canal filling (Figure 1f,g).

**Follow-up**

At a clinical evaluation 12 months later, the tooth was restored with glass-ionomer cement. The tooth was asymptomatic, and the periodontal pocket depth did not exceed 3 mm.

A CBCT radiographic inspection at the same time revealed advanced bone healing (Figure 1).

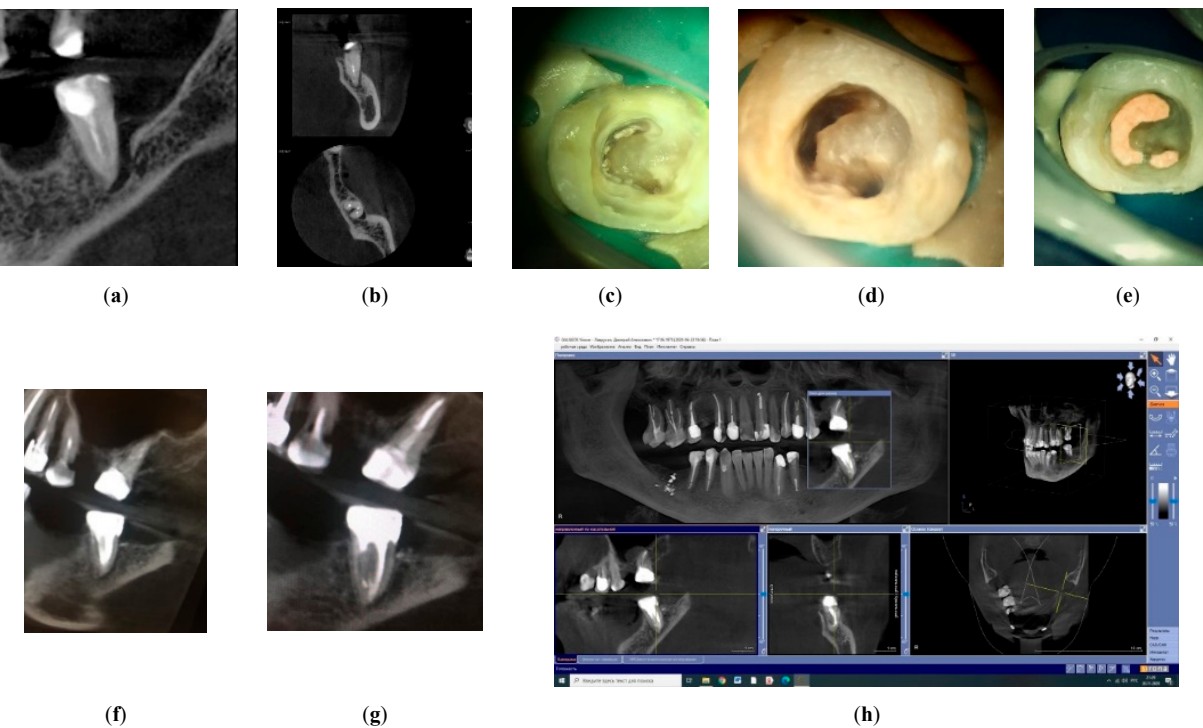

(**a**)    (**b**)    (**c**)    (**d**)    (**e**)

(**f**)    (**g**)    (**h**)

**Figure 1.** Clinical case 1: (**a**,**b**) preoperative CBCT demonstrates radiolucent lesion on the distal aspect of the second left mandibular molar; (**c**) clinical view of the access cavity, remnants of obturation material; (**d**) clinical view of the access cavity after instrumentation of the root canals; (**e**) clinical view of the access cavity after obturation of root canal treatment with GuttaFlow Bioseal; (**f**,**g**) postoperative CBCT demonstrates complete root canal filling; and (**h**) a follow-up CBCT (after 12 months) demonstrates healing.

### 2.2. Clinical Case 2

A healthy 30-year-old Caucasian woman presented at the clinic with a chief complaint of pain in the first maxillary right molar.

**Intra-oral clinical examination**

The tooth was restored with a composite resin restoration, there was tenderness on percussion, and a sinus tract with purulent exudate was present on the buccal surface. The periodontal pocket depth was 8–10 mm buccally, and the periodontal pocket width was 3–4 mm.

**Radiographic examination**

Preoperative CBCT imaging revealed previous root canal obturation and an extensive lesion in the furcation area, with loss of the cortical plate at the buccal surface. In addition, there was a radiolucent lesion in the apical area, and a perforation of the floor of the tooth cavity was suspected (Figure 2i).

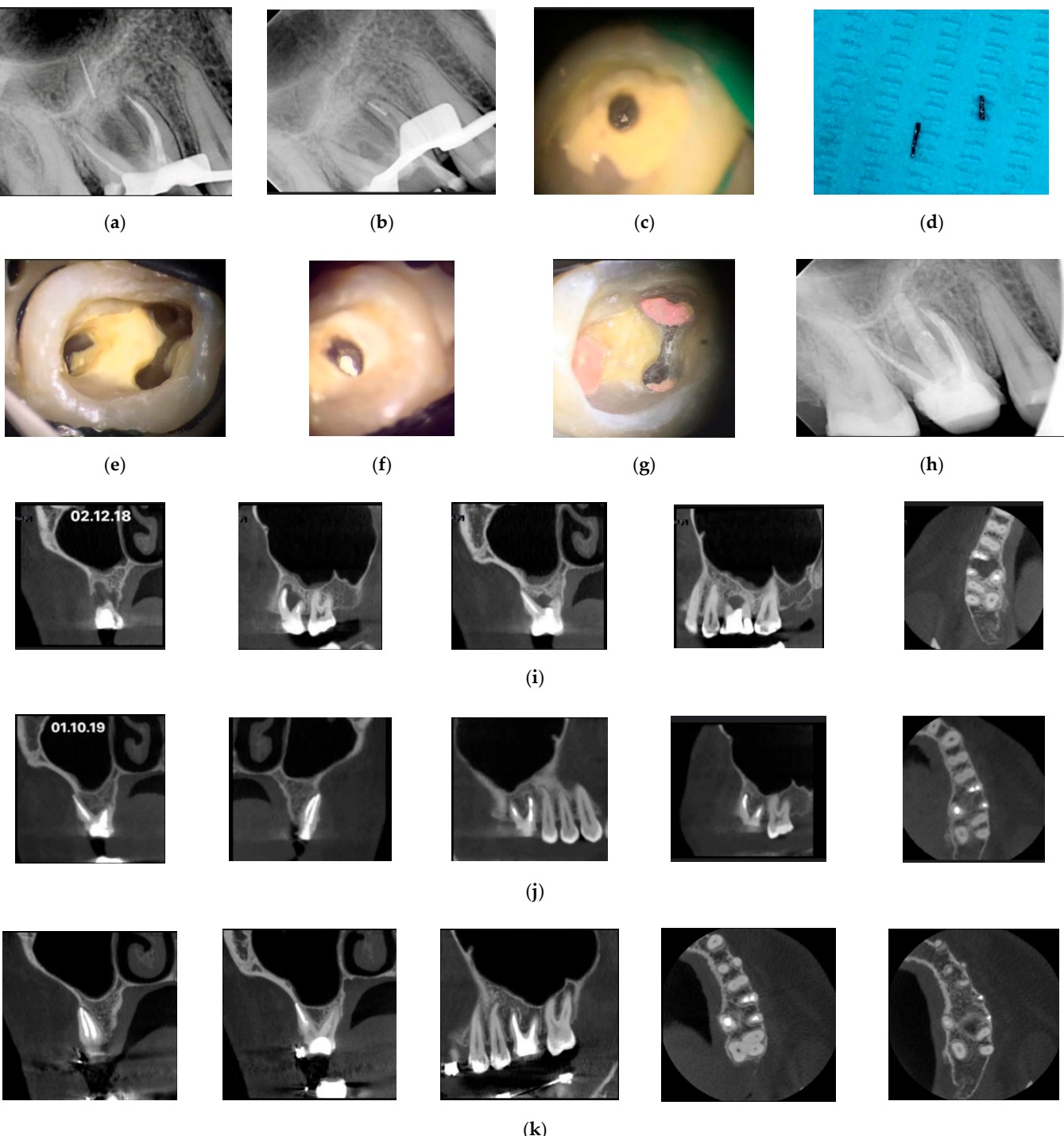

**Figure 2.** (**a**,**b**) Preoperative periapical radiographs of the first upper molar, presenting the "bone" defect as a radiolucent area with irregular contour in the apical region and in the bifurcation region. (**c**,**d**) Stages of removal of fragments of endodontic instruments from the buccal medial and palatal canals. (**e**) Tooth appearance after instrumentation of the root canals. (**f**) Introduction of BioDentin into the palatal canal. (**g**) Root canal obturation. (**h**) Postoperative periapical radiograph. (**i**) Diagnostic CBCT. (**j**) Follow-up CBCT 11 months later. (**k**) Follow-up CBCT 24 months later.

On periapical radiograph examination, a radiolucent area in the furcation was observed, which extended to the apical region of the palatal root (Figure 2a). Fragments of endodontic separated instruments were observed in the palatal and mesio-buccal canals (Figure 2b).

**Endodontic Diagnosis**

Pulpal diagnosis—previously treated

Periapical diagnosis—chronic apical abscess

Primary endodontic—secondary periodontal lesion

Tooth perforation

**Periodontal Diagnosis**

Localized periodontitis, Stage III

**Treatment**

After a discussion of the treatment options with the patient, a decision was made in favor of endodontic retreatment followed by periodontal treatment.

Treatment under magnification using an operating microscope required two visits.

At the first visit, following local anesthesia (2% lidocaine with 1:100,000 epinephrine), the tooth was isolated using a rubber dam, the old restoration was removed, and an appropriate access cavity was refined. On access, perforation was observed in the furcation area, and this was sealed with Bio-Dentin (Septodont, Figure 2c).

At the second visit, 2 days later, retreatment was carried out using the Re-mover 30/07 instrument (MicroMega). The separated instruments were removed using an RT system (EMS) and abundant irrigation with 5% sodium hypochlorite solution (Figure 2d).

The canals were then prepared according to the HyFlex EDM protocol, with intermediate active irrigation using 5% sodium hypochlorite solution with an endoactivator and 17% EDTA solution (Figure 2e).

The buccal root canals were obturated with the GuttaFlow Bioseal cold flowing gutta-percha system (Coltene, Roeko, Germany) using the single cone technique (Figure 2g). Due to significant resorption of the apical region of the palatal canal, the apical third was filled with BioDentin (Septodont, Figure 2f) [4], followed by GuttaFlow Bioseal (Coltene, Roeko, Germany). Postoperative radiographs showed complete root canal filling (Figure 2h). A temporary filling with glass-ionomer cement was applied. Periodontal therapy followed the completion of the endodontic treatment.

**Follow-up**

At a clinical evaluation 11 months later, the tooth was restored with a monolithic zirconia crown. The tooth was asymptomatic, and the periodontal pocket depth did not exceed 3 mm.

A follow-up CBCT radiographic inspection at the same time revealed almost complete restoration of the bone tissue in the periapical region and in the bifurcation region (Figure 2j).

An additional CBCT radiographic evaluation after 24 months revealed advanced periapical healing (Figure 2k).

*2.3. Clinical Case 3*

A healthy 30-year-old Caucasian woman presented at the clinic with a chief complaint of discomfort in the first right upper molar.

On intra-oral clinical examination, the tooth was asymptomatic. The tooth was restored with a composite resin restoration, and the periodontal pocket depth was 6–8 mm.

**Radiographic examination**

Preoperative CBCT imaging revealed previous root canal obturation, with a radiolucent area in the periapical region of all the roots, and mucosal thickening of the maxillary sinus membrane (Figure 3a,b).

**Endodontic Diagnosis**

Pulpal diagnosis—previously treated

Periapical diagnosis—symptomatic apical periodontitis

Primary endodontic—secondary periodontal lesion

**Periodontal Diagnosis**

Localized periodontitis, Stage III

#### Treatment

After a discussion with the patient about the treatment options, a decision was made in favor of endodontic treatment followed by periodontal treatment.

Treatment was performed under magnification using an operating microscope. After local anesthesia (2% lidocaine with 1:100,000 epinephrine), the tooth was isolated using a rubber dam, and an endodontic access cavity was refined. The root canals were prepared and disinfected with the Remover 30/07 instrument (MicroMega, Coltene, Roeko, Germany). During the instrumentation process, the canals were irrigated with 5% sodium hypochlorite solution with an endoactivator and 17% EDTA solution. A total of seven root canals were identified (Figure 3d,e). The canals were instrumented according to the HyFlex EDM protocol.

In the buccal canals, obturation was performed with GuttaFlow Bioseal (Coltene, Roeko, Germany) (Figure 3f). Due to significant resorption of the apical region, the apical third of the palatal canal was filled with BioDentin (Dentsply, Switzerland, Ballaigues), followed by GuttaFlow Bioseal. A temporary filling with glass-ionomer cement was applied. Periodontal therapy was applied after completion of the endodontic treatment.

#### Follow-up

At a clinical evaluation (after 11 months), the tooth was restored with a monolithic zirconia crown. At this time, the tooth was asymptomatic, and the periodontal pocket depth did not exceed 3 mm.

CBCT radiography at the same time revealed healing of the periapical lesion (Figure 3g–l). No pathological changes were observed in the maxillary sinus area.

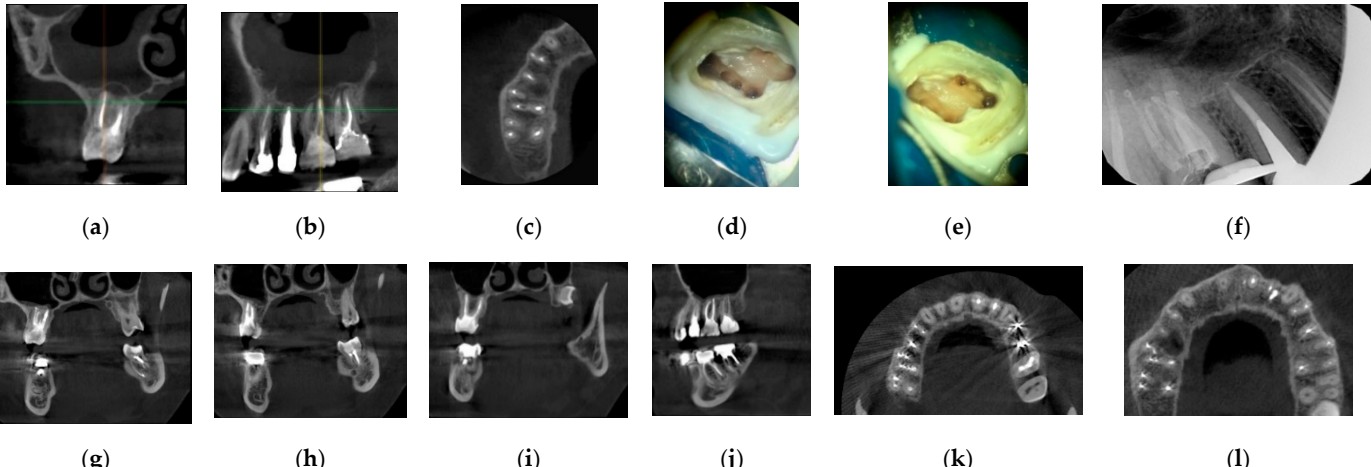

**Figure 3.** Clinical case 3. (**a**) CBCT (coronal section) with a periapical radiolucent lesion and mucosal thickening of the right maxillary sinus membrane. (**b**) CBCT (lateral section) with a periapical radiolucent lesion and mucosal thickening of the right maxillary sinus membrane. (**c**) CBCT (axial section) with a radiolucent area visible on the first right maxillary molar. (**d**,**e**) Tooth appearance after instrumentation of the root canals. (**f**) Final periapical radiographic after the retreatment. (**g**–**i**) CBCT follow-up after 11 months (coronary section). (**j**) CBCT follow-up after 11 months (lateral section). (**k**,**l**) CBCT follow-up after 11 months (axial section).

## 3. Discussion

This article presents three clinical cases representing combined apical and marginal pathology cases that were resolved successfully with endodontic and periodontal treatment.

The comprehensive multidisciplinary approach described here is of utmost importance in the diagnosis and management of endodontic-periodontal lesions in order to maximize the chance of achieving an optimal outcome.

Correct endodontic diagnosis based on dental history, and clinical and radiographic evaluation play important roles in prognosis of the endodontic-periodontal lesions and in the preparation of an optimized treatment plan.

GuttaFlow Bioseal, a bioactive root canal filling material, was used in all of the cases. GuttaFlow Bioseal is composed of gutta percha, polydimethylsiloxane, platinum catalyzer, zirconium dioxide, and bioglass [10].

The biological activity of bioglass is provided by the high content of $Na_2O$ and $CaO$, and a relatively high level of $CaO/P_2O_5$ Hench [2]. The key element is silicon, and the activity is due to the formation of a chemical bond between the surface of the bioglass particles and the surrounding tissue [11]. Hydrolysis of the bioglass in the interstitial fluid leads to the formation of a thin jelly-like layer (gel) of silicic acid—$SiO_2xH_2O$. Negatively charged hydroxyl groups on the surface of the silicic acid layer attract $Ca^{2+}$ ions from the surrounding interstitial fluid solution; then, once the surface charge becomes positive, phosphate ions are deposited on the surface and amorphous calcium phosphate deposits are formed. This leads to the crystallization of hydroxyapatite and the formation of a layer of apatite. As the next stage, biologically active molecules from the extracellular matrix are adsorbed on this apatite layer, and these promote the migration of macrophages and stem cells to the damaged area. Thus, the differentiation of stem cells, and the formation and crystallization of bone matrix occur on the surface of bioactive glass [12].

In dentistry, bioceramics based on calcium phosphates are used to restore bone defects and based on calcium silicates and bio-aggregates (mineral trioxide aggregate, MTA) to ensure the process of apexification and filling of endodontic perforations [11,13,14].

HyFlex EDM NiTi files was used for the instrumentation. Preservation of the original anatomy of the root canal during its preparation is extremely important [12,15].

HyFlex EDM NiTi files are manufactured using the electric discharge machine method. They have a controlled shape memory and, at the same time, possess a unique combination of high flexibility and resistance to breakage [16]. This is because the thermal action of pulsed electrical discharges, excited between the electrode tool and the work piece, means that the entire surface of the file is active and not just the edges, as in tools made by the classical milling method [17]. Another important property is that the phase composition of HyFlex EDM files allows them to change form by reorientating the component martensite, which increases their resistance to torsion loads compared with other files, including those made using wire technology [18].

The CBCT imaging used here provides comprehensive information about the localization of the resorption defect in relation to periodontal tissues, teeth, and nearby anatomical structures (maxillary sinus, mandibular canal, etc., recommendation of the European Society of Endodontics) [19,20].

It should be noted that, as with any device emitting ionizing radiation, the use of CBCT is justified only if the benefits significantly outweigh the risks. In order to avoid mistakes in diagnosis and to avoid harm to the patient, it is important to consider the ALARA principle ("as low as reasonably achievable"). In accordance with the ALARA concept, the diagnostic effect should be higher than the risk of radiation-induced diseases [21].

Some studies have also suggested the use of calcium hydroxide paste over several visits for endodontic regeneration [15]. However, when choosing such a treatment option, care should be taken to avoid possible mechanical and chemical irritation of the periodontal tissues when introducing material into the defect area because this worsens the prognosis [22].

## 4. Conclusions

We presented three clinical cases using a modern endodontic technique including the use of a bioactive root canal sealer to successfully treat endodontic-periodontal disease. No conclusions can be made from this limited number of cases. However, it indicates that further research using these materials and methods would be beneficial.

**Author Contributions:** In the preparation of this manuscript the following contributions were made: conceptualization and literature research, D.L. and I.T.; writing—original draft preparation, D.L., I.T. and B.R.S.; writing—review and editing the final manuscript, I.T., B.R.S., V.V.R., V.A.K., I.S.B., R.A.P. and D.L. All authors have read and agreed to the published version of the manuscript.

**Funding:** This research received no external funding.

**Informed Consent Statement:** Informed consent was obtained from all subjects involved in the study.

**Data Availability Statement:** Not applicable.

**Conflicts of Interest:** The authors declare no conflict of interest.

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
