# Peer review of "Endodontic Management of Endo-Perio Lesions"

_applsci, doi:10.3390/app112311293_

Round 1
Reviewer 1 Report
The issue of endo-perio lesions is indeed a serious issue which might often result in tooth loss. This is why, managing to correctly treat such cases is a great success. The present article provides some information on the topic but much more must be provided in order for it to present the required scientific value to make it publishable.
When an endo-perio lesion is diagnosed, both endodontic and periodontal therapy 66 may be required. In this case, the tooth prognosis depends mainly on the success of the 67 periodontal treatment, assuming that the endodontic procedures are adequate. (6,7) 68 The aim of this study was to present clinical cases of endo-perio lesions that were 69 successfully treated using a modern endodontic technique. ------ If the prognosis depends mostly on the periodontal treatment, why is only the endodontic technique studied in this article?
Thus, we conclude that an adequate endodontic intervention with subsequent obtura- 281 tion of the root canals with a bioceramic sealer is the optimal method to ensure success- 282 ful treatment of combined apical and marginal lesions of the periodontium in one or two 283 visits, often without subsequent surgical procedures. (12-15) ------- Little information is given throughout the entire article on the periodontal treatment, although its importance is clear in treating this case. Three cases is much too little to conclude that this therapeutic option is optimal.
Please provide more comparisons between the results of the present study and other published papers. Also, a clearer protocol describing all the treatments applied to the teeth (endo and perio) in the case reports must be described.
Author Response
1. Materials and Methods: The methods and materials employed in the endodontic treatment are explained in detail. 2. Results Presentation: The results of the three cases presented are well-described using periapical radiographs, CBCT images as well as description of both the initial and follow-up clinical examinations. 3. Conclusion validation: The conclusion as originally described in the article was invalid. The primary cause of healing was the appropriate endodontic treatment, not the periodontal treatment , which was secondary and minimal. The cases presented were primary endodontic, secondary periodontal disease cases. As well, with only the three cases submitted, no conclusions can be made at all. At the most, only the suggestion that treating further cases using similar materials and methods may be reasonable, perhaps adding to a larger database from which additional research may be performed. 4. The manuscript was revised and resubmited.
Reviewer 2 Report
Dear Authors,
I think the article could be accepted after the suggested changes.

Author Response

(The authors gave the same response as above.)

Round 2
Reviewer 1 Report
The issues addressed in the previous review were addressed.
The information provided in this article is somehow relevant to the field.